# The Impact of Liquid Biopsies Positive for *EGFR* Mutations on Overall Survival in Non-Small Cell Lung Cancer Patients

**DOI:** 10.3390/diagnostics13142347

**Published:** 2023-07-12

**Authors:** Jonnathan Roldan Ruiz, Marta Gracia Fuentes Gago, Luis Miguel Chinchilla Tabora, Idalia Gonzalez Morais, José María Sayagués, Mar Abad Hernández, Maria Rosa Cordovilla Pérez, Maria Dolores Ludeña de la Cruz, Edel del Barco Morillo, Marta Rodriguez Gonzalez

**Affiliations:** 1Department of Clinical Oncology, Biomedical Research Institute of Salamanca (IBSAL), University Hospital of Salamanca, University of Salamanca, 37007 Salamanca, Spain; jroldanr@saludcastillayleon.es (J.R.R.); ebarco@saludcastillayleon.es (E.d.B.M.); 2Department of Thoracic Surgery, University Hospital of Salamanca, 37007 Salamanca, Spain; mfuentesgago@saludcastillayleon.es; 3Department of Pathology, Biomedical Research Institute of Salamanca (IBSAL), University Hospital of Salamanca, 37007 Salamanca, Spain; lmchinchilla@saludcastillayleon.es (L.M.C.T.); aidalia_dsks@hotmail.com (I.G.M.); ppmari@usal.es (J.M.S.); marabad@usal.es (M.A.H.); mdludena@saludcastillayleon.es (M.D.L.d.l.C.); 4Department of Pulmonology, University Hospital of Salamanca, 37007 Salamanca, Spain; rcordovilla@saludcastillayleon.es

**Keywords:** non-small lung cancer, liquid biopsy, precision medicine, overall survival

## Abstract

In recent years, non-small cell lung cancer treatment has been revolutionized. *EGFR* tyrosine kinase inhibitors and our improved understanding of its alterations have driven new diagnostic strategies. Liquid biopsies have emerged as a useful tool in these contexts, showing potential utility in early diagnosis combined with low-dose CT scans, as well as potential in monitoring treatment response and predicting the development of patients. We studied the circulating tumor DNA (ctDNA) of 38 *EGFR*-mutated non-small cell lung cancer patients at diagnosis in different moments of their disease by liquid biopsy techniques. Our results show that mean overall survival was significantly lower when a liquid biopsy was positive for the detection of *EGFR* mutations compared with wild-type patients in their liquid biopsy in both univariate (29 ± 4 vs. 104 ± 19 months; *p* = 0.004) and multivariate analysis (*p* = 0.008). Taking this into consideration, liquid biopsies could be key to improving the control of this disease.

## 1. Introduction

Lung cancer is the leading cause of death from cancer worldwide, representing 11.4% of new diagnoses and accounting for 18% of all cancer deaths [1,2,3]. There are two main types of lung cancer: small cell lung cancer (SCLC) and non-small cell lung cancer (NSCLC). About 85% of lung cancers are NSCLCs, which encompasses different histological subtypes such as adenocarcinomas, squamous cell carcinomas, and large cell carcinomas [4]. Up to 80–90% of patients with NSCLC are symptomatic at the time of diagnosis, showing frequent respiratory symptoms related to primary tumors. Patients may also present symptoms associated with metastasis, particularly neurological symptoms because of the involvement of the central nervous system, and bone pain [4]. The main risk factor for the development of NSCLC continues to be tobacco. Although, over the last decades, tobacco consumption rates have fallen among the global population, in Spain, tobacco consumption remains at a high level among men aged 15–64 years [5,6]. Furthermore, an increase in NSCLC cases among non-smoking women has been observed, meaning that the underlying causes are unclear (e.g., they may be related to hormonal exposure, previous radiotherapy for the treatment of other malignancies such as breast cancer, home environmental factors, etc.) [7,8].

The treatment of patients with lung cancer depends on the morphology, tumor stage, molecular characteristics, and assessment of a patient’s overall medical (OS) condition. Surgery is the most effective treatment, but it is reserved for patients with stage I, II, or IIIA NSCLC (representing only 35% of total cases) [4]. Since most NSCLC patients present an advanced-stage disease at diagnosis, systemic treatment with cytotoxic agents is often used as the standard treatment, showing a modest survival benefit [9].

A better understanding of the underlying molecular mechanisms that encourage tumorigenesis in NSCLC has led to the development of many target tools aimed at specific genetic abnormalities. The discovery of activating mutations of the Epidermal Growth Factor Receptor (*EGFR*) gene, as well as the advancement of *EGFR* tyrosine kinase inhibitors (TKI), has had a significant impact on adaptive treatment strategies and, consequently, on survival among patients with advanced NSCLC [10]. In addition to these TKI acting against EGFR mutations, in recent years, there has been a genuine revolution in the treatment of lung cancer through the development of drugs that act against *ALK*, *ROS1*, and *RET* rearrangements [11,12,13], *BRAF* [14,15,16], and *KRAS* mutations [17], as well as the standardization immunotherapy based on the immunohistochemical expression of PD-L1 [18,19]. Therefore, NSCLC has brought personalized medicine to the fore globally, and the most recently published guidelines propose that advanced non-squamous NSCLC and selected patients with squamous cell carcinomas (SCCs) should undergo genetic testing [20,21].

Despite these advances on the onset of the disease, monitoring patients for early relapse, and detecting resistance to new targeted therapies, challenges continue to persist. Diagnoses and histological typification are carried out by examining a small piece of tissue sample or even through cytology, with a likely lack of sufficient material for a proper molecular diagnosis or without the possibility to obtain a new tumor sample [22,23,24,25,26].

A liquid biopsy (LB) is a minimally invasive alternative to surgical biopsies that is based on analyzing circulating tumor cells (CTCs), cell-free DNA in plasma (cfDNA), circulating miRNA, exosomes, and tumor-educated platelets (TEP) in a biological fluid (mainly blood) [27]. These methods have emerged as a powerful tool to identify molecular alterations in patients with cancer. In addition to the practically non-invasive nature of this procedure, other advantages of liquid biopsies include their rapid processing and optimal reflections of tumor heterogeneity. They are also highly useful both after ablative treatment and in monitoring the response to targeted therapy. The major constraint is the very low amount of ctDNA. At present, there are various strategies and platforms available; however, thus far, there is no clear consensus on when or when not to perform a liquid biopsy [26,28,29].

Over the past few years at our hospital, a referral hospital in the mid-west of Spain, we have performed LBs and examined *EGFR* mutational status in accordance with the Medical Oncology Service’s guidelines; in most cases, patients have an evolution that is worse than expected. The objectives of the present study were to analyze our hospital’s clinical management of *EGFR*-mutated NSCLC patients, with a specific emphasis on how we used LB techniques, and assess the impact of LB techniques on tumor evolution and OS among patients.

## 2. Materials and Methods

### 2.1. Study Design

This is a retrospective observational study based on the first 79 liquid biopsies of patients treated at the Pathological Anatomy Service of the Complejo Asistencial Universitario de Salamanca between October 2016 and December 2022. Patients without *EGFR* alterations in the biopsy were excluded; consequently, this meant that our cohort consisted of 38 patients. All of them had non-squamous NSCLC diagnosed via small biopsy specimen or fine needle aspiration (FNA). Clinical (age, sex, symptoms, manifestations, type of sample for diagnosis, stage, surgery), histological (diagnosis, TTF1 and PDL1, staining), and evolutionary (treatment, recurrence, death) variables were collected. The biomarkers studied in the clinical treatment of NSCLC (*ALK*, *ROS1*, and *BRAF* genes) did not show relevance in the present study (Appendix A).

### 2.2. Tissue Biopsy Procedure

Tissue samples were obtained from surgical or biopsy specimens and were formalin-fixed paraffin-embedded (FFPE) tissue blocks. A 3 μm section stained with hematoxylin-eosin was made from each of the samples. Immunohistochemistry techniques (TTF1 (Leica, Microsystem Ltd., Milton Keynes, UK)) and p40 staining (A. Menarini Diagnostics, San Diego, CA, USA, a 1:200 dilution) were used for all the samples via the use of an automated immunostainer (Bond Polymer Refine Detection, Leica Microsystem Ltd., Milton Keynes, UK) and following the manufacturer’s instructions. Nuclear staining was considered positive in >10% of tumor cells. Furthermore, in 4 of these cases, histochemical techniques, such as periodic acid–Schiff (PAS) and periodic acid–Schiff–diastase (PAS diastase) for glycogen detection with the Alcian Blue/Periodic Acid-Schiff Stain kit on Agilent ArtisanLink platform from DAKO (Stevens Creek Blvd. Santa Clara, CA, USA), were also performed. A real-time PCR test was performed to examine the most common mutations in exons 18, 19, 20, and 21 of *EGFR* gene (Cobas^®^ 4800 EGFR mutation test v2) on purified DNA samples that had been isolated from the tissue samples derived from biopsies.

### 2.3. Liquid Biopsy Procedure

When it was not possible to perform a re-biopsy and in cases where the oncologist needed updated patient information, data on at least one prior liquid biopsy were obtained per patient during the course of their disease (Appendix A). Peripheral blood samples (usually around 10 mL) were collected in Vacutainer EDTA tubes (BD, Plymouth, UK) for each liquid biopsy. For the first hour, the plasma was subjected to high-speed centrifugation (10,000 rpm for 10 min at 4 °C), and immediately after that, a second centrifugation (16,000 rpm for 10 min at 4 °C) was performed to pellet cell debris. After centrifugation, at least 2 mL of the supernatant was aliquoted into 2 mL cryotubes and stored at −80 °C. The commercially available Cobas^®^ cfDNA sample preparation kit was then used to extract ctDNA. DNA quantification was carried out by using a NanoDrop^®^ ND-1000 spectrophotometer (ND-1000, NanoDrop Technologies, Wilmington, DE, USA). We used the FDA-approved LB test for detecting EGFR mutations (Cobas^®^ EGFR Mutation Test v2 for blood/from plasma samples) (CE-IVD) (Roche Molecular Diagnostics, Branchburg, NJ, USA).

### 2.4. Statistical Analysis

All the clinical/biological, morphological, genetic, and evolutionary variables were collected in a database, and the statistical program SPSS v.21 (IBM Corp., Armonk, NY, USA) was used to calculate the statistical significance of the different variables, as well as OS. For normally distributed continuous variables, we used Student’s *t*-test, and for non-normally distributed ones, the Mann–Whitney U test was used. OS curves were determined according to the Kaplan–Meier test. Multivariate analyses of prognostic factors influencing OS were performed using Cox regression, only taking into a count the variables that showed a significant association with OS in the univariate analysis; a *p* value of <0.05 (or Pearson-corrected *p*, if applicable) indicated statistical significance.

## 3. Results

Of the 38 patients included in the present study, 17 were men (45%) and 21 women (55%); the median age of 68 years (43–85 years). The majority (25; 66%) were non-smokers (Table 1). Functional status was assessed at two levels. Although it is true that the 38 patients presented good general status, 24 of them were classified as 0 (66%) according to the Eastern Cooperative Oncology Group’s (ECOG) guidelines, and the remaining 14 were classified as ECOG 1 (34%). Notably, 18% of patients (seven cases) were completely asymptomatic, and their diagnosis was incidental. This observation was made before complementary control or monitoring tests were carried out for the other non-tumor pathologies. A total of 22 patients (58%) presented symptoms associated with the primary tumor, the majority of which were manifested via coughing, expectoration, hemoptysis, dyspnea, or chest pain; 9 (24%) patients showed symptoms related to the location of the metastases, highlighting neurological manifestations, and pain at different levels of the spine. Nearly three quarters of the patients had stage IV diseases (28 cases, 74%); five patients had a locally advanced stage III level of disease (13%), and a small minority presented early-stage diseases (three with stage I, accounting for 8% of patients, and two with stage II, accounting for of patients 5%). Most of the patients did not undergo surgery (33; 87%); therefore, their histological diagnosis was obtained, in most cases, via a small biopsy (transbronchial biopsy, core-needle biopsy (CNB) of the lung, fine needle aspiration (FNA), or a cytological examination based on pleural fluid analysis).

Among the samples received (36 cases), 95% were diagnosed as adenocarcinomas, while two (5%) were deemed undifferentiated NSCLC (Table 2). On the one hand, p40 was entirely negative in all cases. On the other hand, 91% (35 patients) were TTF1 positive, and we performed PAS and PAS diastase in two of the three negative cases (9%), with positive results being obtained for both samples.

In the tissue samples, del(19) and L858R mutations were found to be the most frequently observed mutations (*n* = 16 cases; 42%), followed by Ins(20) (*n* = 3; 8%) and the combined mutations del(19) + L861Q (*n* = 1; 3%), del(19) + T790M (*n* = 1; 3%), and G719X + S578I (*n* = 1; 3%) (Appendix A). Of the 38 patients, 32 (84%) started first-line treatment with first-generation tyrosine kinases inhibitors (TKIs), 20 with Erlonitib, and 10 with Gefitinib; 2 patients were assigned chemotherapy as their standard for of treatment to manage mutational mutation, specifically Ins(20) resistance. In addition, four patients (10%) received chemotherapy after post-surgical recurrence.

At least one liquid biopsy was performed to study *EGFR* mutations in all patients in the cohort (Appendix A), and the median follow-up was 24 months (4–154 months). All LBs were performed at the request of the Medical Oncology Service of our hospital. As shown in Figure 1, in 71% of the patients (*n* = 27), the presence of *EGFR* mutations was detected in at least one LB, with del(19) and L858R being the most frequent mutations (10 cases; 26%), followed by the combined mutations del(19) + T790M (4 cases; 10%). During patient monitoring, 13 patients developed T790M mutation resistance, leading to a change in their therapeutic approach to incorporate Osimertinib, a third-generation TKI with sensitivity to this mutation. Only one patient presented this alteration at diagnosis (associated with del(19)). In these 13 patients, the median onset of the T790M mutation from diagnosis was 34 months (7–78 months).

As expected, positivity for TTF1 was observed in most cases (*n* = 36; 95%), while expression in tumor cells was detected in half of the cases studied, with expression not being associated with LB positivity (Table 2).

Finally, the mean OS among patients was 48 ± 10 months. During patient monitoring, 31 of the 38 patients died (82%), showing statistically significant differences between patients with positive and negative LBs (27 cases (71%) vs. 11 cases (29%), respectively, *p* = 0.004). From a prognostic point of view, the variables that showed a significant influence on OS in the univariate analysis were treatment, ECOG, and LB (*p* < 0.05) (Table 3). However, in the multivariate analysis, only LB maintained significance (HR = 4.7; *p* = 0.008), with a mean OS of 29 ± 4 vs. 104 ± 20 months in positive and negative LBs, respectively (*p* 0.004) (Figure 2).

## 4. Discussion

*EGFR* mutations have become a very important therapeutic target in patients with NSCLC, especially in advanced stages. They have a significant impact on both OS and quality of life [30]. Therefore, improving our knowledge of these alterations, as well as the role that LBs play in detecting them and in the monitoring of these patients, is essential for improving therapeutic strategies and adapting treatment regimes so that align with patient’s stage of disease. The prevalence of lung cancer has led to recent developments in personalized medicine and revolutionized patient management by forcing clinical institutions to integrate clinical data, pathological findings, and molecular profiling.

The results obtained in our study are consistent with the literature regarding the profile of the patients studied. Overall, 60% of our patients were women, and 67% declared themselves to be non-smokers [31,32]. This confirms the increasing trend of NSCLC cases among women, especially non-smokers [7,8]. Even though tobacco use rates have generally decreased over time among men, they continue to be high [5,33,34]. In fact, half of the male patients (*n* = 8) in our study were smokers or ex-smokers, and we only found this habit in 23% of the female patients (6 out of 26), a trend that is seemingly consistent with what is published in the literature [3,34,35,36,37]. Likewise, it is estimated that 53% of lung cancer cases are diagnosed between the ages of 55 and 74 [34], with the median age of diagnosis in our study being 68 years. It should be noted that the median age of diagnosis in women is approximately 8 years older than in men (65 ± 13 years vs. 72 ± 10 years), probably due to decreased direct exposure to tobacco, although the etiological factors of lung cancer not associated with smoking are yet to be defined (e.g., hormonal exposure, previous radiotherapy due to other neoplasms such as breast cancer or environmental factors) [7,36,38,39,40] and are not well-characterized.

Even though surgery is the best form of treatment for patients with early-stage NSCLC, approximately 35–55% of patients relapse during the first 5 years after surgery [41,42,43,44]. The causes of this circumstance have not been adequately elucidated, but some authors have pointed to the percentage of solid components in the lesion, the histological subtype, the micropapillary growth pattern, necrosis, airborne spread, etc. [45,46,47]. In our series, 5 of the 38 patients were considered to be in the early stage of their disease (I–II), and after surgery, all of them presented local or distant relapses throughout the course of the disease. Nevertheless, up to 80% of patients with lung cancer are diagnosed in advanced stages, where the main diagnostic technique is usually a small biopsy or FNA [48,49,50]. This frequency was slightly higher in our study as almost 90% of the patients were in a locally advanced stage (*n* = 6; 14%) or metastatic (*n* = 32; 74%), confirming the fact that lung cancer is often diagnosed in the advanced stages, when therapeutic options are limited. Because of this reason, detection systems that allow for early stages diagnoses and therefore increase the possibility of curative treatment strategies are being extensively researched. In this regard, studies have been carried out to demonstrate the benefit of low-dose computed tomography (LDCT) in patients with risk factors [51]. However, this approximation method has some limitations, examples of which include false positives (with consequent overdiagnosis) and the radiation dose administered to patients who are healthy a priori [28]. In order to solve those problems, the value of the LB as an auxiliary tool to LDCT is being explored both as a screening method and as a diagnostic method in the early stages of the disease through molecular identification in peripheral blood [42,52]. Indeed, TRACERx has shown that ctDNA can be detected between 6 and 12 months before the presence of radiologically detectable lesions, which, in a selected population, could complement LDCT, decreasing the number of diagnostic interventions [53].

Previous reports based on the study of the mutational status of the *EGFR* gene, both in tissue and fluid biopsies, show that the mutations del(19) and L858R, followed by ins(20), are the most frequent mutations observed in NSCLC patients [54]. In line with other studies that used similar methodological approaches to that of this study, our results showed the presence of del(19) and L858R in almost half of the NSCLC cases studied. We only observed a complex mutation in two patients (G719X + S578I, and del(19) + L681Q). Complex *EGFR* mutations comprise a heterogenous group of mutations with a prevalence between 5 and 15% of all *EGFR* mutations [54]. In alignment with these observations, in our study, complex mutations were observed in the 6% of cases. The presence of the T790M mutation in the diagnosis of the disease is infrequent. Indeed, it has been suggested that it is a germline mutation of the *EGFR* gene [55] and has been associated with a poor prognosis [54,56,57,58]. In our study, only one patient presented a mutation associated with del(19) at diagnosis, and no significantly lower median survival was observed compared to other patients with different mutations (19 vs. 20 months).

Even though routine LB tests are not recommended in the diagnosis and follow-up of NSCLC [59], the results obtained in our study, although small, are strikingly significant. We observed that, in patients in whom the EGFR mutation was not detected in LB during follow-up, overall survival was four times higher than patients with at least one *EGFR* mutation detected during their LB (mean OS of 29 ± 4 vs. 104 ± 19 months; *p* 0.001, respectively). These data suggest that LB could have a powerful predictive value in the outcome of patients with NSCLC. Most of the cases with positive LB were detected between the first and the second determination, which implies that two ordinary blood extractions could infer the evolution of the patient. In our study, we used quantitative PCR (Cobas EGFR mutation test V2); however, there are currently more sensitive techniques and technologies available, such as microfluidic technology (digital PCR [60] and BEAMing [61]) or massive parallel sequencing (NGS) [62], which could further increase the predictive value of the mutational status of the EGFR gene. In the FASTACT-2 study [63], a baseline LB was performed, and on the first day of the third cycle, they observed the presence of *EGFR* mutations in the second LB, associating it with a negative impact on overall and disease-free survival. Similarly, a LUNGCA-1 study observed that the detection of EGFR mutations through the LBs of operated patients can predict relapses [64]. Although it is true that there is no consensus on when to perform a LB, aside from when searching for resistant mutations (T790M to first- or second-generation EGFR-TKI; C797S to third generation EGFR-TKI) or after an observed decreased in response to treatment, our own findings reinforce the idea of incorporating it into the routine of oncological control [65]. In this way, a LB could be performed at baseline even before starting treatment and/or in the postoperative period and every 3 months during the first year, when resistant mutations usually appear [10,63,66,67,68].

In 2019, the addition of Osimertinib to the first-line treatment of advanced NSCLC [69] decreased interest in searching for the T790M resistant mutation in newly diagnosed patients. In our study, LBs detected this mutation in 13 cases (34%) throughout the follow-up period; thus, it is possible that LBs could enable shifts to more effective treatment strategies, positively impacting quality of life and survival [63]. At present, the mutations demonstrating resistance to Osimertinib mechanisms are already known, such as the loss of the T790M mutation (early event) or C797S mutation, which is present in 7% of cases [54], and more information on these mechanisms will be made available over time thanks to new NGS techniques

## 5. Conclusions

In summary, our results show that the presence of mutations in the *EGFR* gene detected using LB techniques in patients with *EGFR*-mutated NSCLC is an independent prognostic factor for OS. Liquid biopsies make it possible to monitor the mutations of a tumor in a relatively simple and quick way, which, together with other diagnostic tools, could be useful in our daily clinical practice, improve prognostic assessments, and guide clinical decision making regarding the treatment of NSCLC patients. Additional prospective studies with larger cohorts are required to validate the utility of LB techniques in the diagnosis and follow-up of NSCLC patients.

## Figures and Tables

**Figure 1 diagnostics-13-02347-f001:**
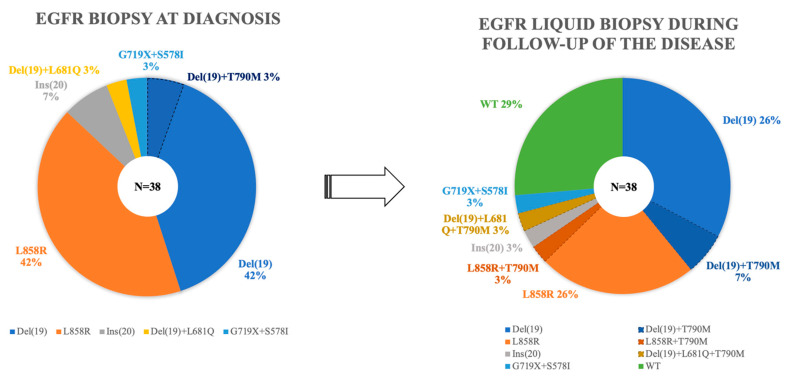
Frequency and type of mutations detected in the *EGFR* gene in the plasma (liquid biopsy) samples from 38 *EGFR*-mutated non-small cell lung cancer (NSCLC) patients.

**Figure 2 diagnostics-13-02347-f002:**
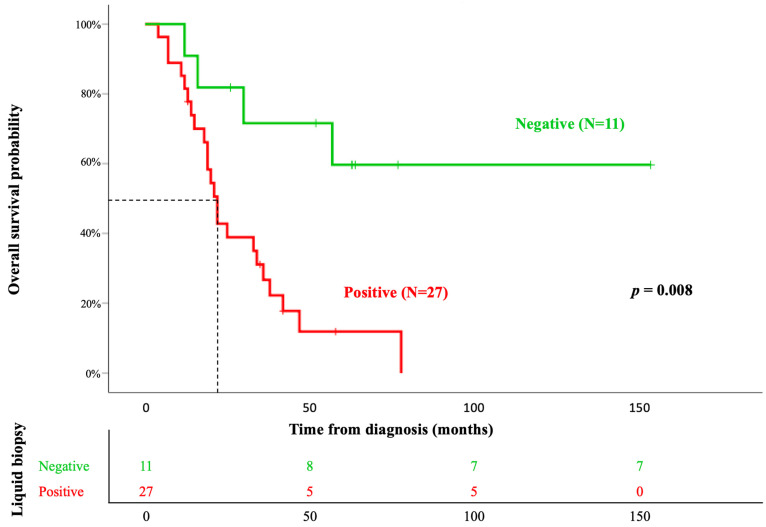
Identification of at least one mutation in the *EGFR* gene in the plasma of *EGFR*-mutated non-small cell lung cancer (NSCLC) patients during disease follow-up showed a significant impact on OS in univariate (*p* = 0.004) and multivariate (*p* = 0.008) analyses.

**Table 1 diagnostics-13-02347-t001:** Clinical and biological characteristics of *EGFR*-mutated non-small cell lung cancer (NSCLC) patients (*n* = 38) according to the identification of *EGFR* mutations detected by using liquid biopsy techniques.

Characteristics	Number of Patients [*n* (%)] *EGFR* Mutations Detected by Liquid Biopsy		Total of Cases (*n* = 38)	*p*
	No (*n* = 11)	Yes (*n* = 27)		
Age, years *	68 (52–83)	73 (43–85)	68 (43–85)	NS
Gender				
Male	5 (45)	12 (44)	17 (45)	NS
Female	6 (55)	15 (56)	21 (55)	
Smoking history				
Smoker or ex-smoker	5 (45)	8 (30)	13 (34)	NS
Nonsmoker	6 (55)	19 (70)	25 (66)	
ECOG **				
0	6 (55)	18 (67)	24 (63)	NS
1	5 (45)	9 (33)	14 (34)	
Symptoms				
Pneumological (cough, dyspnea)	4 (36)	18 (67)	22 (58)	NS
Non-pneumological	3 (27)	6 (22)	9 (24)	
Asymptomatic	4 (36)	3 (11)	7 (18)	
AJCC stage ***				
I	0 (0)	3 (11)	3 (8)	0.02
II	2 (18)	0 (0)	2 (5)	
III	3 (27)	2 (18)	5 (13)	
IV	6 (55)	22 (81)	28 (74)	
Surgery				
Yes	1 (9)	4 (15)	5 (13)	NS
No	10 (91)	23 (85)	33 (87)	
Oncologic Treatment				
None	1 (9)	0 (0)	1 (3)	
Chemotherapy	3 (27)	1 (4)	4 (10)	0.01
Radiotherapy	1 (9)	0 (0)	1 (3)	
Tyrosine kinase inhibitor	6 (55)	26 (96)	32 (84)	
Exitus				
Yes	4 (33)	23 (85)	27 (29)	0.002
No	7 (67)	4 (15)	11 (71)	
OS (months)	104 ± 19	29 ± 4	48 ± 10	0.001

Results are expressed as the number of cases (percentage) or * as the median (range). ** ECOG: general status according to the Eastern Cooperative Oncology Group. *** AJCC indicates American Joint Committee on Cancer (based on 8th ed. of the AJCC). NS: statistically nonsignificant (*p* > 0.05).

**Table 2 diagnostics-13-02347-t002:** Histopathological and expression profile of *EGFR*-mutated non-small cell lung cancer (NSCC) patients (*n* = 38) according to the identification of *EGFR* mutations detected by using liquid biopsy techniques.

Characteristics	Number of Patients [*n* (%)] *EGFR* Mutations Detected by Liquid Biopsy		Total of Cases (*n* = 38)	*p*
	No (*n* = 11)	Yes (*n* = 27)		
Histology				
Adenocarcinoma	10 (91)	26 (97)	36 (95)	NS
Undifferentiated carcinoma	1 (9)	1 (3)	2 (5)	
TTF1 expression				
Positive	10 (91)	25 (93)	35 (91)	NS
Negative	1 (9)	2 (7)	3 (9)	
PD-L1 expression *				
Negative (0%)	4 (50)	12 (48)	16 (48)	NS
Positive	4 (50)	13 (52)	17 (52)	

Results are expressed as the number of cases (percentage). * Performed in 33 cases. NS: statistically nonsignificant (*p* > 0.05).

**Table 3 diagnostics-13-02347-t003:** Clinical, biological, and genetic characteristics of *EGFR*-mutated non-small cell lung cancer (NSCLC) patients (*n* = 38) and their association with OS.

Variable	*N*	Univariate Analysis	Multivariate Analysis	HR (95% CI)
Age				
<68 years	15 (39)	NS		
≥68 years	23 (61)			
Gender				
Male	17 (45)	0.1	NS	
Female	21 (55)			
Smoking history				
Smoker or ex-smoker	13 (34)			
Nonsmoker	25 (66)	NS		
ECOG *				
0	24 (63)			
1	14 (37)	0.05	NS	
AJCC stage				
I	3 (7)			
II	2 (5)	NS		
III	5 (13)			
IV	28 (74)			
Surgery				
No	33 (87)	NS		
Yes	5 (13)			
Oncologic treatment				
None	1 (3)			
Chemotherapy	4 (10)	0.08	NS	
Radiotherapy	1 (3)			
Tyrosine kinase inhibitor	32 (84)			
TTF1expression				
Positive	35 (92)	NS		
Negative	3 (8)			
PD-L1expression				
Positive (≥1%)	17 (52)	NS		
Negative	16 (48)			
*EGFR* status on liquid biopsy				
Positive	27 (71)	0.004	0.008	4.7 (1.5–12.9)
Negative	11 (29)			

Results are expressed as the number of cases (percentage). NS: statistically nonsignificant (*p* > 0.05). * ECOG: general status according to the Eastern Cooperative Oncology Group.

## Data Availability

All study data can be viewed in the manuscript.

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
