# Peer review of "The Impact of Liquid Biopsies Positive for EGFR Mutations on Overall Survival in Non-Small Cell Lung Cancer Patients"

_diagnostics, 2023, doi:10.3390/diagnostics13142347_

Round 1

Reviewer 1 Report

This article by Roldan Ruiz et al. describes a retrospective survival analysis of patients with NSCLC in relation to the results of liquid biopsy for the detection of EGFR mutations. An association was found between a positive result in liquid biopsy and a lower overall survival.

The study is interesting and provides insights into the usefulness of liquid biopsy for EGFR mutations in the prognostic of NSCLC patients. These data would support the prognostic value of liquid biopsy testing in the treatment of NSCLC.

Minor points:

  • In the running title, the abbreviation lacks a letter: NS_LC

  • On lines 36-37, some words are in italics (...NSCLC encompassing different histological subtypes as adenocarcinoma, squamous cell carcinoma, and large cell carcinoma [4]. Up to 80-90% of patients with…). Is there any reason for that?

  • The spanish word “y” remains on line 104 (The immunohistochemistry techniques (TTF1 (Leica, Microsystem Ltd, Milton Keynes UK) y p40 (A. Menarini Diagnostics, San Diego, USA, a 1:200 dilution)).

  • BL abbreviation is used instead of LB on lines 197, 214, 270, 273, 274, 282, 287, and 303.

Major points:

  • The authors state in the title of the article and the legend to Figure 2 that liquid biopsy has an impact on OS in patients with NSCLC. This claim could be misleading, as it implies an effect of the liquid biopsy procedure on patient survival, whereas the study results and conclusions only suggest an association between a positive liquid biopsy test result and decreased overall survival. I would ask the authors to qualify their assertion.

  • The use of subheadings (eg, Study Design, Patients, Tissue Biopsy Procedure, Liquid Biopsy Procedure, EGFR Contact Detection, Statistical Analysis) in the Materials and Methods section would be recommended. It would provide order and readability to the section.

  • On lines 237-238 it is stated that 5 of 43 patients were at stage I-II of disease. This number don’t coincide with the total number of patients in the study cohort.

  • In the Summary it is stated that the median OS in both study groups is 29±4 vs. 104±19 months. These figures appear in Table 1 along with a median overall OS of cases of 48±10 months. However, on line 193 it is stated that the median OS for the series was 30±6 months. Moreover, on line 199 the figures 29±4 vs.104±19 months are referred to as mean, not median. These discrepancies must be resolved.

  • The authors state in lines 98-99 that the biomarkers that are studied in the clinical routine of NSCLC (genes KRAS, BRAF, ALK, and ROS1) did not show relevance in the present study and are therefore not shown. Does it mean that there were no statistically significant differences between the studied groups (positive vs. negative liquid biopsy result)? To the extent that mutations in these genes are known to have an impact on the prognosis of NSCLC, either one must demonstrate that there are no real differences between groups or include these data in the analysis.

  • Although Kaplan Maier analysis seems to be properly done, a better description in the statistical methods is necessary. Especially, it is necessary to justify the assumptions to apply Cox regression (proportional hazards assumption, in this case).

  • The p indicated in Figure 2 is 0.004, which corresponds to univariate regression analysis for EGFR liquid biopsy status. However, it would be more correct to show p for the multivariate analysis (0.008), even though both are reflected in the legend of the figure.

The English used in the article has no grammatical problems and is generally correct. However, the use of long and complex sentences makes it not very fluent in some passages. A review of the language of some points of expression would be advisable.

Author Response

General Comment. - The study is interesting and provides insights into the usefulness of liquid biopsy for EGFR mutations in the prognostic of NSCLC patients. These data would support the prognostic value of liquid biopsy testing in the treatment of NSCLC.

Minor points:

Comment 1.- In the running title, the abbreviation lacks a letter: NS_LC

Answer to comment 1.- We thank the reviewer for pointing out this mistake that has now been corrected.

Comment 2.- On lines 36-37, some words are in italics (...NSCLC encompassing different histological subtypes as adenocarcinoma, squamous cell carcinoma, and large cell carcinoma [4]. Up to 80-90% of patients with…). Is there any reason for that?

Answer to comment 2.- The italics have been removed.

Comment 3.- The spanish word “y” remains on line 104 (The immunohistochemistry techniques (TTF1 (Leica, Microsystem Ltd, Milton Keynes UK) y p40 (A. Menarini Diagnostics, San Diego, USA, a 1:200 dilution)).

Answer to comment 3.- Thanks again to the reviewer for pointing out the bug which we have now fixed

Comment 4.- BL abbreviation is used instead of LB on lines 197, 214, 270, 273, 274, 282, 287, and 303.

Answer to comment 4.- The manuscript was carefully revised and the misspellings corrected, including the change of the abbreviation BL to LB.

Major Points

Comment 1.- The authors state in the title of the article and the legend to Figure 2 that liquid biopsy has an impact on OS in patients with NSCLC. This claim could be misleading, as it implies an effect of the liquid biopsy procedure on patient survival, whereas the study results and conclusions only suggest an association between a positive liquid biopsy test result and decreased overall survival. I would ask the authors to qualify their assertion.

Answer to comment 1- The title and legend of Figure 2 of the manuscript have been modified according to the reviewer's comments.

Comment 2.- The use of subheadings (eg, Study Design, Patients, Tissue Biopsy Procedure, Liquid Biopsy Procedure, EGFR Contact Detection, Statistical Analysis) in the Materials and Methods section would be recommended. It would provide order and readability to the section.

Answer to comment 2.- Following the reviewer's suggestion, the material and methods section of the revised manuscript has been structured in 4 subheadings (Patients, Tissue Biopsy Procedure, Liquid Biopsy Procedure and Statistical Analysis)

Comment 3.- On lines 237-238 it is stated that 5 of 43 patients were at stage I-II of disease. This number don’t coincide with the total number of patients in the study cohort.

Answer to comment 3.- We appreciate the reviewer bringing our attention to this error, which has been corrected.

Comment 4.- In the Summary it is stated that the median OS in both study groups is 29±4 vs. 104±19 months. These figures appear in Table 1 along with a median overall OS of cases of 48±10 months. However, on line 193 it is stated that the median OS for the series was 30±6 months. Moreover, on line 199 the figures 29±4 vs.104±19 months are referred to as mean, not median. These discrepancies must be resolved.

Answer to comment 4.- The discrepancies pointed out by the reviewer have been resolved in the new version of the revised manuscript.

Comment 5.- The authors state in lines 98-99 that the biomarkers that are studied in the clinical routine of NSCLC (genes KRAS, BRAF, ALK, and ROS1) did not show relevance in the present study and are therefore not shown. Does it mean that there were no statistically significant differences between the studied groups (positive vs. negative liquid biopsy result)? To the extent that mutations in these genes are known to have an impact on the prognosis of NSCLC, either one must demonstrate that there are no real differences between groups or include these data in the analysis.

Answer to comment 5.- Following the indication of this Reviewer, we have now included a new table with the data regarding the mutational status / rearrangements of these genes in the new version of the revised manuscript (Supplementary Table 1).

Comment 6.- Although Kaplan Maier analysis seems to be properly done, a better description in the statistical methods is necessary. Especially, it is necessary to justify the assumptions to apply Cox regression (proportional hazards assumption, in this case).

Answer to comment 6:  We fully agree with the reviewer. Following his/her comment, the paragraph of the materials and methods section of the revised version of the manuscript has been re-written carefully to further explain how we conducted the study to address our objectives.

Comment 7.- The p indicated in Figure 2 is 0.004, which corresponds to univariate regression analysis for EGFR liquid biopsy status. However, it would be more correct to show p for the multivariate analysis (0.008), even though both are reflected in the legend of the figure.

Answer to comment 7: We completely concur with the reviewer. The p indicated in Figure 2 is 0.004 has replaced to 0.008, explaining it in the legend of the Figure.

Comments on the Quality of English Language

The English used in the article has no grammatical problems and is generally correct. However, the use of long and complex sentences makes it not very fluent in some passages. A review of the language of some points of expression would be advisable.

Answer to comment on the quality of English language: The language of the manuscript has been carefully revised by a native English Professor, following the recommendation of the reviewer.

Reviewer 2 Report

In this work, Ruiz et al. present a compelling case for the use of liquid biopsy as a potentially revolutionary tool in the diagnosis and management of NSCLC patients. The focus on EGFR tyrosine kinase inhibitors as a significant therapeutic strategy is particularly topical, given its growing significance in NSCLC treatment. Overall The authors have also made a good attempt to discuss the implications of their findings, making a persuasive argument for the increased utilization of liquid biopsy in the clinical management of NSCLC patients. I believe the manuscript can be accepted in its current form.

Author Response

General Comment. - In this work, Ruiz et al. present a compelling case for the use of liquid biopsy as a potentially revolutionary tool in the diagnosis and management of NSCLC patients. The focus on EGFR tyrosine kinase inhibitors as a significant therapeutic strategy is particularly topical, given its growing significance in NSCLC treatment. Overall The authors have also made a good attempt to discuss the implications of their findings, making a persuasive argument for the increased utilization of liquid biopsy in the clinical management of NSCLC patients. I believe the manuscript can be accepted in its current form.

Answer to General Comment:  We thank the reviewer for the overall positive comments about the paper and the work contained in it.

Round 2

Reviewer 1 Report

The authors have

The authors have fixed all the points I raised in my review report. I congratulate them for their work.